# Stochastic properties of musical time series

Corentin Nelias[1] & Theo Geisel[1,2] ✉

Musical sequences are correlated dynamical processes that may differ depending on musical styles. We aim to quantify the correlations through power spectral analysis of pitch sequences in a large corpus of musical compositions as well as improvised performances. Using a multitaper method we extend the power spectral estimates down to the smallest possible frequencies optimizing the tradeoff between bias and variance. The power spectral densities reveal a characteristic behavior; they typically follow inverse power laws ($1/f^{\beta}$-noise), yet only down to a cutoff frequency, where they end in a plateau. Correspondingly the pitch autocorrelation function exhibits slow power law decays only up to a cutoff time, beyond which the correlations vanish. We determine cutoff times between 4 and 100 quarter note units for the compositions and improvisations of the corpus, serving as a measure for the degree of persistence and predictability in music. The histogram of exponents $\beta$ for the power law regimes has a pronounced peak near $\beta = 1$ for classical compositions, but is much broader for jazz improvisations.

Music can be viewed as a correlated dynamical process with a succession of pitches, chords, rhythmic values, etc. The nature of the correlations is related to the degree of expectation and surprise in the musical progression, which was emphasized by music theorists since Leonard Meyer[1,2]. The emotional power of music and musical meaning are thought to depend on the interplay between anticipation and uncertainty. It is difficult to substantiate these concepts by quantitative approaches. An obvious approach is to try to quantify this interplay on the basis of information theoretical concepts, e.g., by estimating redundancies and entropies of compositions in order to compare different composers, musical genres, etc. Entropies can be estimated for distributions of single pitches (unigrams) and pairs of successive notes (digrams)[3–5]. For larger n-grams (sequences of notes), however, the finite length of musical compositions imposes severe limitations[4]; the combinatorial explosion of the number of possible states precludes accurate estimates of their probabilities, as the length of the time series is fixed. In particular the ultimate goal of determining block entropies and redundancies for sufficiently large n-grams (e.g., representing entire musical phrases) in a composition remains an unaccomplishable challenge.

Understanding music as a correlated dynamical process raises the question what is the nature of these correlations and makes music a subject for complex systems research. With the growing interest in nonlinear dynamics, new techniques for time-series analysis were developed[6–10] and it is natural to apply them also to musical time series. Considering the difficulties in determining block entropies and redundancies, it is more promising and feasible to characterize musical sequences by their correlation decay. The autocorrelation function can quantify how much an event at time $t = \tau$ is related to an event at $t = 0$ and thus in its decay can also reflect the degree of persistence or the degree of innovation. It is convenient to determine its properties through estimates of the power spectral density (PSD), which according to the Wiener-Khinchin theorem is related to the autocorrelation function by Fourier transform[11]. For musical compositions and melodies the focus of time-series analysis is naturally on pitch sequences (pitches are particularly well adapted for time-series analysis, as they possess an intrinsic ordering and can be represented in dependence on time.). The power spectral analysis of such sequences of pitches is unrelated of course to the series of harmonics or overtones of single tones, but reflects the stochastic properties of the progression of successive pitches on much longer time scales, desirably up to the length of musical movements.

In early work, Voss and Clarke analyzed PSDs of full audio signals of recordings and reported $1/f$-noise ("pink noise") in loudness fluctuations and frequency fluctuations, i.e. $f^{-1}$-decays of the PSD[12]. Boon and Decroly studied pitch sequences and in contrast reported $1/f^{2}$-noise, also known as "red noise" i.e., $f^{-2}$-decays of the PSD[13]. Nettheim, on the other hand, found power-law decays of the melody-PSD which

[1]Max Planck Institute for Dynamics and Self-Organization, 37077 Göttingen, Germany. [2]Bernstein Center for Computational Neuroscience Göttingen, Georg August University Göttingen, 37073 Göttingen, Germany. ✉e-mail: theo.geisel@ds.mpg.de

follow $1/f^\beta$-shapes with $\beta \in [1, 2]$ and with a tendency towards $1/f^2$-shapes[14]. Unfortunately, due to the necessary averaging, the lowest frequencies in his melodic spectra correspond to times of up to 4 bars only, and thus reflect correlations only within musical phrases and not in entire melodies. Other authors have used detrended fluctuation analysis (DFA), which yields fluctuation functions $F(s)$ that are also related to autocorrelation functions and PSDs. Applying this to the time series of pitch frequencies in Bach's inventions, Jafari et al.[15] reported power law exponents for $F(s)$, which typically correspond to $f^{-1/2}$ decays of the PSDs (i.e., $\beta$ near 1/2), whereas González-Espinoza et al.[16], who analyzed 304 music scores of different composers, found that the fluctuation function $F(s)$ could not always be described by a single power law, but followed various profiles, e.g., with two different power law exponents in different regimes.

These different and partly contradicting observations lead to quite different conclusions concerning the properties of the correlation decay. For PSDs decaying like $1/f^\beta$ with $0 < \beta < 1$ the autocorrelation function asymptotically decays in a power law $t^{-\gamma}$ with $\gamma = 1 - \beta$. If a $1/f$-decay (i.e. with $\beta$ close to 1) is observed down to the smallest frequencies, it implies a very slow power-law decay of correlations and scale-free behavior on arbitrarily long timescales. This represents a very far reaching conclusion, which has generated most of the interest in the problem. If, however, the PSD diverges like $f^{-1/2}$ for $f \to 0$, the autocorrelation function asymptotically decays like $t^{-1/2}$ instead. An observed $1/f^2$-decay on the other hand may also be associated with a fast exponential decay of correlations (when the spectrum follows a Lorentzian).

Among these claims, the most spectacular is the report of $1/f$-noise by Voss and Clarke[12]. If confirmed by PSD-analyses of real pitch sequences down to smallest frequencies, it would imply that musical compositions still keep a small but finite memory of earlier events after arbitrarily long times. Voss and Clarke have reported $1/f$-decays over 4 orders of magnitude down to very low frequencies corresponding to hours of recording.

Their observation, however, does not translate into pitch autocorrelations, since in their analysis of audio signals of entire recordings they only could consider a vague measure of momentary sound frequency, the number of successive zero crossings of the audio signal per unit time. It is not clear what this observation implies for musical pitch sequences.

The observations of such diverse power laws outlined above, which were based on different techniques and different musical material, cannot be reconciled and thus the important question remains open, what are the long-time autocorrelation properties of pitch sequences in musical compositions. A main obstacle and possible cause of differing observations is the need to reserve part of the time series for averaging in order to achieve reliable PSD-estimates. Therefore only a fraction of the entire time series can be used and its length typically determines the bandwidth or smallest possible frequency of the estimate. Is there a way, nevertheless, to answer this fundamental question by careful statistical analyses going down to small frequency scales and up to long timescales? Are the PSDs of pitches characterized by power laws on sufficiently many scales? If yes, on which frequency scales do they occur and what are their typical exponents?

To answer these questions, in the present work we carry out careful estimates of the PSD of pitch sequences in a large corpus of musical compositions as well as in improvised performances. As mentioned above, reliable estimates require sufficient averaging and thus sufficiently long time series are needed. Using a multitaper PSD-method we strive to extend the PSD-estimates down to the smallest possible frequencies (given by the bandwidth) and to optimize the tradeoff between bias and variance. In each considered example we explicitly determine the bandwidth and the confidence intervals to indicate the range of validity of our conclusions.

Based on such careful estimates we find that the PSD of musical pitch sequences typically follows inverse power laws only down to a cutoff frequency, where the PSD turns into a flat plateau. The cutoff frequency thus marks a transition from $1/f^\beta$ to a white noise behavior at low frequencies. Correspondingly the pitch autocorrelation function exhibits slow power law decays only up to a cutoff time, beyond which the correlation vanishes. This transition from a strongly correlated to an uncorrelated time series at large time differences may reflect the interplay between predictability and uncertainty, between expectation and surprise in music as discussed by music theorists. The cutoff time can serve as a measure for the degree of persistence and predictability in musical compositions. It can be used to characterize different compositions and in some cases shows differences between composers - we find, e.g., that on average it seems to be larger in Mozart's compositions than in Bach's. In general we obtain cutoff times in a range between 4 quarter note units and 100 quarter note units; they show an increasing trend in dependence on the lengths of the compositions.

In the power law regime of the PSD we determine the histogram of the power law exponents $\beta$ across all pieces, which has a pronounced peak near $\beta = 1$ for classical compositions. So one can speak of $1/f$-noise in musical pitch sequences (and of very slowly decaying correlations), yet only in a limited frequency range bounded by the cutoff frequency. In some cases, however, we find pure power laws without plateaus and cutoff frequencies. This is the case only for relatively short musical time series, where the PSD bandwidth is necessarily relatively large. Therefore it is very likely that in these cases a true cutoff is hidden at a smaller frequency within the bandwidth and would show up, if the bandwidth could be lowered. The spectra also exhibit "rhythmic" peaks at multiples and fractions of inverse quarter note units. They reflect the rhythmic structure that is present in the different pieces.

In improvised jazz solos, pure power laws are very rare and plateaus dominate at small frequencies. The rhythmic peaks are broader and less pronounced indicating a larger timing variability in jazz solos.

## Results

We performed PSD estimations of 553 pitch time-series in total, of which 99 were extracted from classical music scores and 454 from transcriptions of improvised jazz solos, using the multitaper method and following the procedures described in detail in the methods section. In the case of classical music, we made a distinction between single movements and full compositional works (e.g. symphonies) as they might involve different correlation structures. We chose NW = 2 as a basis (see subsection on variance and bias) for our estimations as it ensures access to low frequencies (bandwidth W at least down to 3 orders of magnitude compared to the highest observable frequency) with an acceptable level of variance reduction. Indeed, as the number of tapers $K$ is commonly given by $K = 2NW - 1$, the choice NW = 2 permits the use of 3 tapers, which means 3 times averaging[17].

### Classical music scores

Even though single movements and full compositional works have different length scales and organizational structures, we found that their PSDs behave similarly. We identified two distinct kinds of structures: *power-law decay* and *power-law with plateau* (which, for conciseness, we abbreviate as PL and PL+P, respectively). Figures 1 and 2 present representative examples of these two structures for single movements and for full compositions with several movements. Additional examples can be found in the supplementary information. As shown in the preceding section, it is formally not possible to decide whether a plateau observed below the bandwidth is a true plateau or merely an artifact of the bias. Therefore distinctions between power-laws and power-laws with plateau can be made only down to the resolution limit of the bandwidth. Nevertheless, in cases where we see a plateau above the bandwidth limit, we can be confident that it is real

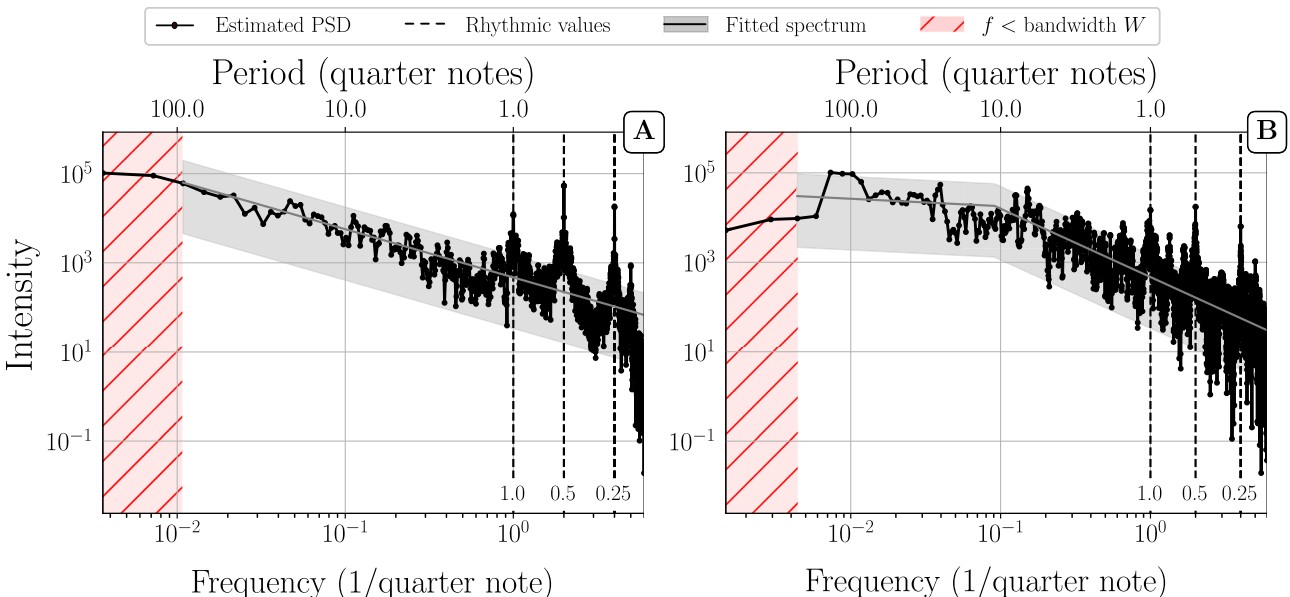

**Fig. 1 | Multitaper PSD estimation of single movements. (A)** Shostakovich's Prelude and Fugue in B op. 87 no. 11 (first violin) and **(B)** Allegretto from Haydn's String Quartet in D major, Hob.III:79 (first violin). Frequency and intensity are shown in a log-log representation and are shown in units of 1/quarter notes. Note, however, that we used a grid of 12 sub-units per quarter notes for the time series segmentation, which yields a highest frequency of 6/quarter note. For convenience, the top horizontal axis shows the time periods in quarter note units corresponding to the frequencies on the horizontal axis. As a grid of 12 sub-units per quarter note was used for the time-series segmentation, the time period is obtained as *Period*(quarter notes) = 1/12*f*. The shaded grey area represents a 95% confidence interval on the basis of the fitted PSD (grey line). The fitting procedure is described in the methods section. The vertical dashed lines mark frequencies corresponding to multiples and subdivisions of quarter notes. The shaded pink area represents frequencies below the bandwidth W. Example **(A)** is representative for a PSD with power-law (PL) down to the bandwidth W, and example **(B)** for a power-law + plateau (PL+P).

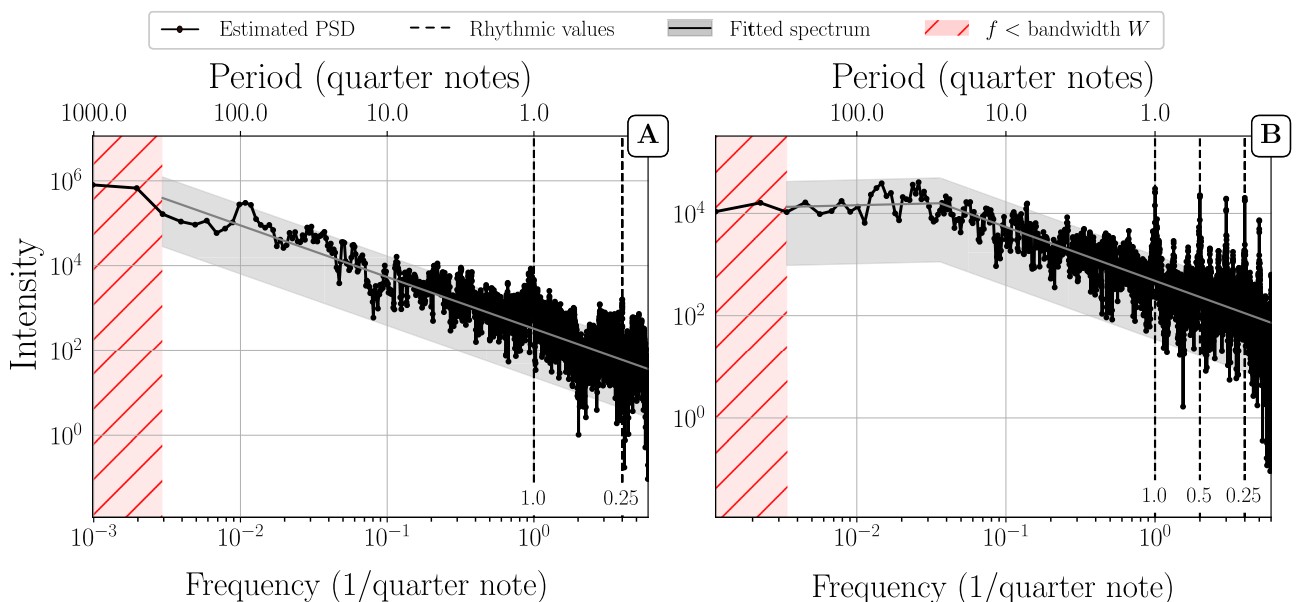

**Fig. 2 | Multitaper PSD estimation of full compositions. (A)** Violin Sonata No.1 in G minor, BWV 1001 (J.S. Bach) and **(B)** Violin Sonata in D major, HWV 371 (G.F. Handel). The shaded grey area represents a 95% confidence interval on the basis of the fitted PSD (grey line). For other conventions, see Fig. 1. Example (A) is representative for a PSD with power-law (PL) down to the bandwidth W, example (B) for a power-law + plateau (PL+P).

and not an artifact (as we outline in the subsection on variance and bias, the bias does not affect flat PSD regions.) and not a numerical artifact. In PL cases without plateau, it is not possible to decide whether a plateau exists, but it is possible that a plateau would show up, could the bandwidth be lowered.

An important implication of the PSDs is on the behavior of the autocorrelation functions, which are related to the PSDs according to the Wiener-Khinchin theorem. If the PSD asymptotically decays in a power law like $1/f^{\beta}$ with $0 < \beta < 1$, the autocorrelation function asymptotically decays in a power law $t^{-\gamma}$ with $\gamma = 1 - \beta$. Such cases represent very slow decays of the autocorrelation function (long-range correlations). If on the other hand the PSD power law ends in a plateau, this implies a relatively abrupt disappearance of autocorrelations above a cutoff time. To yield a qualitative understanding of the effect

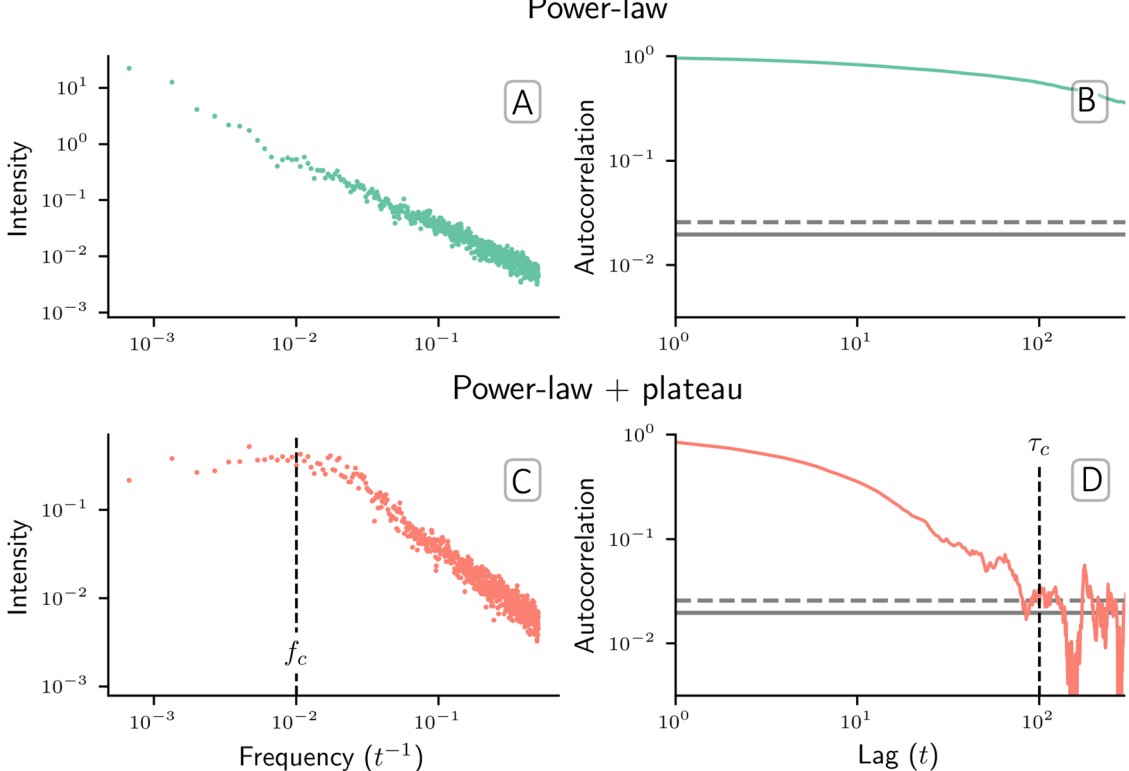

**Fig. 3 | Comparison of autocorrelation functions from simulated signals presenting a PL and a PL+P with a cutoff frequency in their PSDs, resp. A**, **B** show a PSD estimation of pink "$1/f$" simulated noise (dimensionless) and the corresponding autocorrelation function, (**C**, **D**) are the PSD and autocorrelation function of a "power-law + plateau" process. The cutoff frequency $f_c$ and the cutoff time $\tau_c = 1/f_c$ denote the frequency and time at which the plateau starts and the correlations end, respectively. The effect of the plateau in (**C**) is to shorten the range of the correlations, as is shown in (**D**). The dashed and the full horizontal lines provide a 99% and 95% confidence interval for the autocorrelation function, respectively. Note that in (**B**), the power-law decay of the autocorrelation function is only expected asymptotically for $t \to +\infty$.

of plateaus on PSDs, Fig. 3 provides a comparison of the autocorrelation function of two simulated signals exhibiting a power-law PSD and a power-law + plateau PSD, respectively. The signal exhibiting a PL (Fig. 3A, B) was generated with a colored noise algorithm[18], the PL+P signal was obtained by manipulating the Fourier transform of colored noise (flattening the end of the spectrum), and taking the inverse Fourier transform after randomizing the phases. Figure 3 shows that the long-range correlations induced by a PL structure (Fig. 3B) are shorter ranged, if the PSD shows a plateau (Fig. 3D). In fact, the appearance of a plateau at frequency $f_c$ implies the existence of a time-scale $\tau_c = 1/f_c$ that provides an upper bound after which correlations stop being significant (Fig. 3D). Flat PSDs are reminiscent of white-noise and correspond to fluctuations of uncorrelated processes.

In order not to surcharge the main body of the present article, we present other PSD estimations for classical music scores in the supplementary information (see Supplementary Figs. 12 and 13). We notice

that the PL structures show up much less frequently than PL+P structures, as can be seen in Table 1, where we show the count of pieces that follow a PL and a PL+P structure. Moreover, these two shapes are not equally distributed among composers. The time period at which a plateau appears shows some variability (it typically lies between 4 and 100 quarter notes), and varies among composers and pieces. For example the movements composed by Mozart which we analyzed, revealed plateaus starting at time periods of approximately 40 quarter notes, whereas many movements in J.S. Bach's compositions showed plateaus starting at time periods of 10 quarter notes. Pitch autocorrelations thus tend to be more short ranged in J.S. Bach's compositions, whereas they tend to be more persistent and long-ranged in Mozart's compositions (for more and quantitative details, see the supplementary information and supplementary Fig. 8).

Overall, we observe that the longest pieces from our corpus overwhelmingly show PL+P structures. The appearance of a plateau, is strongly related to the total length of the piece, as only shorter pieces tend to present PL shapes. This can be seen in Fig. 4, which shows a histogram representing the number of PL and PL+P pieces as a function of piece length. This histogram only considers single movements, as full compositions almost exclusively exhibit PL+P shapes. We note, in particular, that all movements longer than 1200 quarter notes show PL+P shapes. This indicates a possible maximal extent of correlations in musical pieces and suggests that the pure power-law decays (PL) observed in short pieces are merely a consequence of their relatively large bandwidth W, which prevents plateaus from showing up in the unbiased frequency range (above W). According to this explanation, plateaus would then show up if the bandwidth could be lowered.

**Table 1 | Classification count of observed behaviors**

| Piece type | Shape | |
|---|---|---|
| | PL | PL+P |
| Improvised solos | 14 | 440 |
| Single movements | 15 | 42 |
| Several movements | 3 | 41 |

The table presents the number of pieces showing PL and PL+P shapes for the three different types of pieces studied (single movements, several movements, and improvised solos). Single and several movements both refer to classical compositions. In all three cases, the PL+P shape is observed more frequently than PL, and is even more dominant in improvised jazz solos compared to classical music.

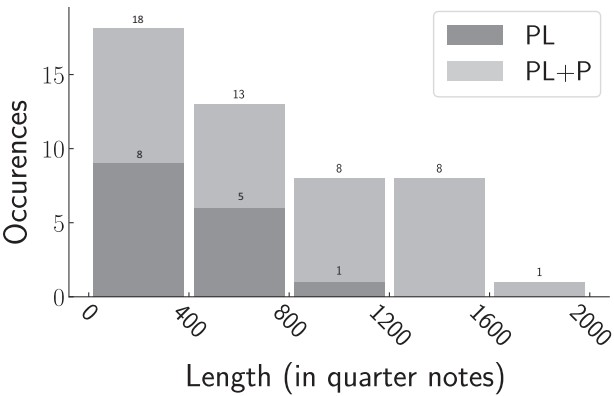

**Fig. 4 | Distribution of PL and PL+P shapes as a function of piece length.** The pieces considered here are single movements. The x-axis shows length bins of equal width. Each bar represents the number of pieces with lengths in between the boundaries of the corresponding bin. Light grey refers to PL+P pieces, dark grey represents PL. Overall, long pieces tend to present a PL+P shape and PL-shapes are absent in movements longer than 1200 quarter notes. This clearly suggests that plateaus do not show up in short pieces, merely because the bandwidth in these pieces is too large to enable the detection of plateaus.

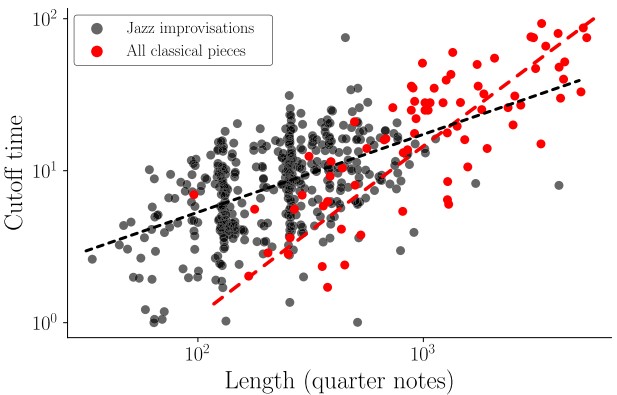

**Fig. 5 | Cutoff time in PL+P pieces as a function of piece length.** The red points show the cutoff times of classical pieces (both single movements and full compositions) presenting a PL+P shape in their PSD estimation as a function of their total length. The black points present the same relationship, for all jazz solos of the Weimar Jazz Database (for details, see next section). For jazz improvisations, we observe a higher density of points around 128 and 256 quarter notes (32 and 64 bars in 4/4 time signature), which are very common durations of solos. For classical compositions, one can see an increasing trend between the lengths of PL+P pieces and the cutoff periods. This trend is somewhat less pronounced for jazz solos.

While the individual composing style of different composers significantly impacts the spectral properties of pitch time-series, one of the most crucial parameters for the overall shape of the PSD seems to be the total length of a piece. Indeed, there exists a correlation between the cutoff period, where the slope of PL+P shapes ends in a plateau, and the length of a piece. We found that plateaus in longer PL+P pieces tend to appear at far lower frequencies than in shorter pieces. In other words, longer PL+P pieces are also associated with larger cutoff periods. This relationship can be seen in Fig. 5, which shows the cutoff period vs piece length for classical pieces and jazz improvisations in log-log representation. The figure displays some scattering, indicating that there are exceptions to the roughly linear trends. We see, however, that the largest cutoff values are only reached for the longest pieces. The trend is more pronounced for classical pieces. A possible explanation for this increasing trend might be that composers

often pick up and vary preceding themes and motifs throughout the whole composition, which tends to cause longer correlations in longer compositions.

In single movements as well as in full compositions, we also noticed the appearance of "rhythmic peaks" on top of the aforementioned structures, which are positioned at common note values (see e.g. Fig. 3). Distinct peaks appear at periods 0.25, 0.5, 1, 2, and 4, which correspond to periods of sixteenth, eighth, quarter, half, and whole notes in musical nomenclature, respectively. These peaks are therefore indicators of the rhythmic structures present in the pieces.

So far we considered the existence and position of cutoff frequencies, but one may ask what are the exponents of the power-law regime in the PSD, as they influence the decay of correlations. The distribution of exponents for the power-law part of the PL+P and PL structures of classical compositions is shown in the histogram of Fig. 6A. For each piece of our corpus, the exponent was determined by fitting the PSD estimation (see Methods section) and isolating the power-law part. The exponents follow a uni-modal distribution centered around 1.1 and spanning the interval [0.3, 1.8]. This result has important consequences for the decay of the autocorrelation function. As pointed out before, if a PSD power law extends down to arbitrarily small frequencies, the autocorrelation then also follows a power law and decays very slowly with an exponent $\gamma = 1 - \beta$. The decay becomes extremely slow, as $\beta$ approaches $\beta = 1$. As we do not find this behavior asymptotically in the PSD in almost all cases, the slow power law decays are only transient in the autocorrelation function and limited by the cutoff times (see also Fig. 3). In this limited sense, bounded by a finite cutoff time, we can speak of long-range correlations in musical pitch time series and correspondingly (with a distribution of $\beta$ values peaking near $\beta \approx 1$) of 1/f-noise in a finite frequency range.

## Jazz improvisations

As classical music compositions typically show more structure and musical form than jazz improvisations, we may expect somewhat different autocorrelation behaviors and consequently different behaviors of the PSDs. We therefore study jazz improvisations as a separate category. Similarly as for the classical music scores, we find PL and PL+P structures (see Fig. 7). For the jazz improvisations, however, pure PL structures are extremely rare (see Table 1) and show up only in short solos, where the large bandwidth W technically precludes the observation of plateaus. The cutoff periods for the crossover from power-law to plateau also show more variability than in classical pieces (Fig. 5). There is a higher density of points around 128 and 256 quarter notes (32 and 64 bars in 4/4 time signature), which represent typical durations of jazz solos.

Since the cutoff period marks a transition from long-range power-law autocorrelations to a white spectrum (lacking autocorrelations) it indicates an upper bound for the time beyond which the (long-range) correlations get lost. The fact that this correlation time is typically smaller for jazz improvisations than for classical compositions reflects a lower degree of musical structure in improvised jazz solos. But note that most of the jazz solos are also much shorter than the lengths of most of the classical music pieces. Perhaps the most notable difference compared to classical compositions is the way, in which rhythmic peaks appear. Since the analyzed solos were live improvisations, the onsets and durations of various notes being played deviate from the "true" value they would have on a corresponding sheet notation. These fluctuations can be intentional[19] and/or result from human error[20]. Depending on their strength and nature they can broaden the rhythmic peaks. Overall, when rhythmic peaks are visible in PSDs of jazz solos, they tend to be less pronounced than in classical compositions. This can be seen, e.g., in Fig. 8A. Looking at both ends of the spectrum we identify a PL+P structure. However, in the intermediate frequency domain, broadened peaks corresponding to multiples of quarter notes form a large bump. Quarter notes are delayed or anticipated by jazz

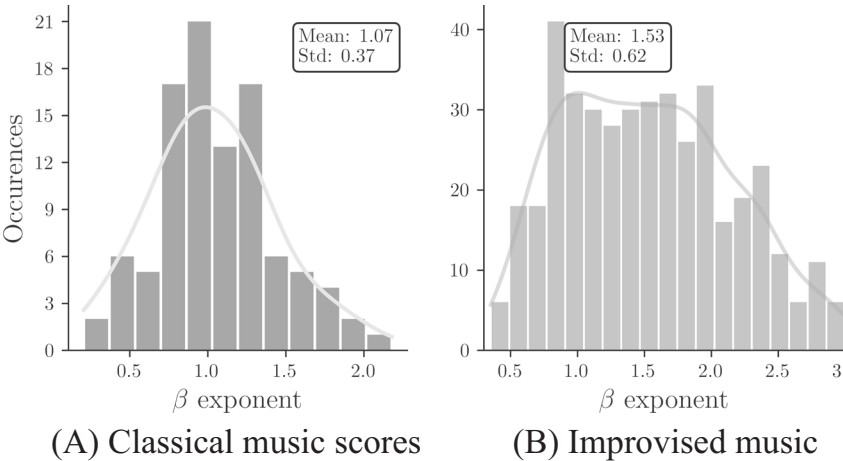

(A) Classical music scores   (B) Improvised music

**Fig. 6 | Histogram representation of β exponent.** The exponents β were extracted from the power law regimes of the PSDs for (**A**) 94 classical music scores from online databases[26–28] (**B**) 454 improvised jazz solos from the WJD[29] based on the high to medium frequency part of piecewise linear fits. The continuous grey line is a guide to the eye obtained from a kernel density estimation. Note that the distribution has a somewhat larger mean and is broader for jazz solos than for classical music scores.

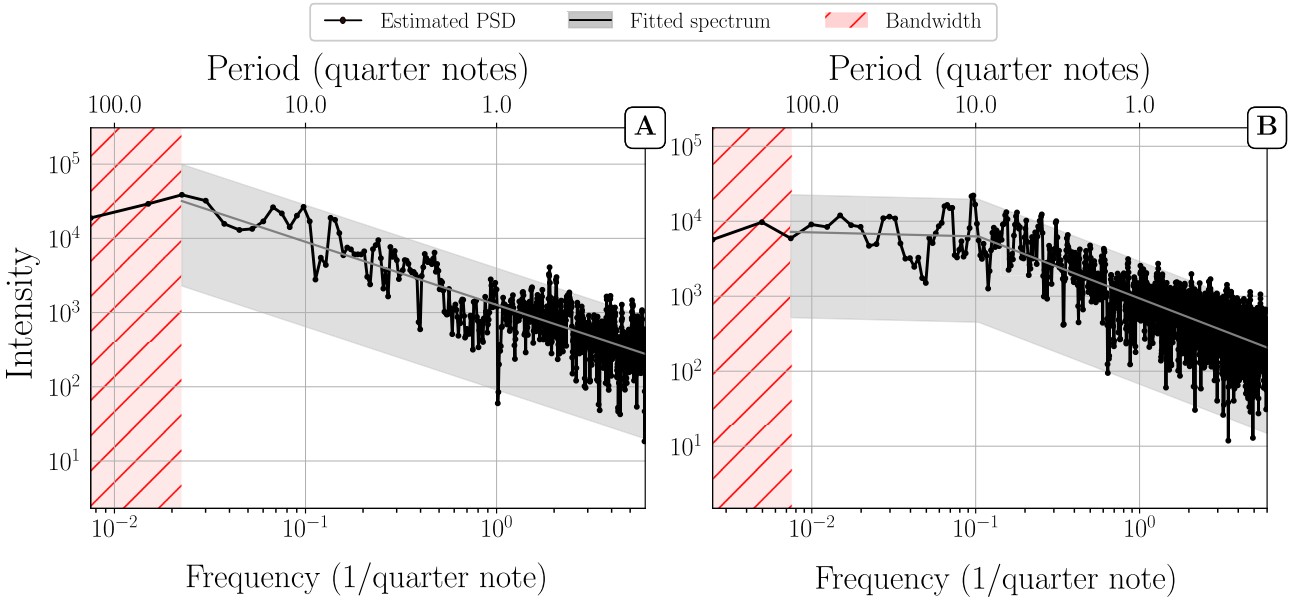

**Fig. 7 | Improvised jazz solos.** Multitaper PSD estimation of (**A**) David Liebman's solo over "Begin the Beguine" and (**B**) Joe Lovano's solo on "I Can't Get Started". The shaded grey area represents a 95% confidence interval based on the fitted PSD (grey line). Jazz solos show a similar PSD structure as classical music, yet with shorter cutoff periods $\tau_c$. Example (**A**) seems to indicate a pure PL shape, but the solo is relatively short such that the large bandwidth W precludes the observation of a plateau.

musicians (e.g., to create surprise and make their solos more interesting) which explains the broadening of peaks at these frequencies. While we occasionally find PL+P shapes in PSDs of improvised solos that show very broad peaks (like in Fig. 8A), the majority of the PL+P structures that we observe closely resemble the ones seen in the case of classical music scores. The rhythmic peaks are less pronounced, however, as exemplified in Fig. 8B.

The distribution of exponents for the power-law part in PL+P and PL structures of improvised jazz solos is shown in the histogram of Fig. 6B. In contrast to what is seen in the case of classical music scores, the distribution shows a broader peak area in interval [−2.0, −0.75] and the mean value of the exponents is larger than in classical music scores.

## Discussion

In the present work, we assessed the correlation structure of musical pitch time series through estimates of their power spectral densities.

We took special care to identify and mitigate sources of bias and variance in the PSD estimation in order to avoid spurious trends. Using a multitaper method we were able to reach the highest possible frequency resolution. In a large corpus of classical music scores and jazz improvisations we generally found the presence of slowly decaying correlations, yet of finite range, as revealed by inverse power-laws in the PSD-estimations that are turning into plateaus at low frequencies. Due to their ubiquity we interpret the appearance of these plateaus as a characteristic of musical time series.

Comparing our results to previous literature[12–16], we find agreement in so far as there is a power law regime, where the power laws do not show a single (universal) exponent. We found a continuum of possible values and in detail have determined the histogram of the power law exponents β. For a large corpus of classical music scores we found a relatively narrow distribution centered near β = 1; for improvised music the distribution was broader. Optimizing the

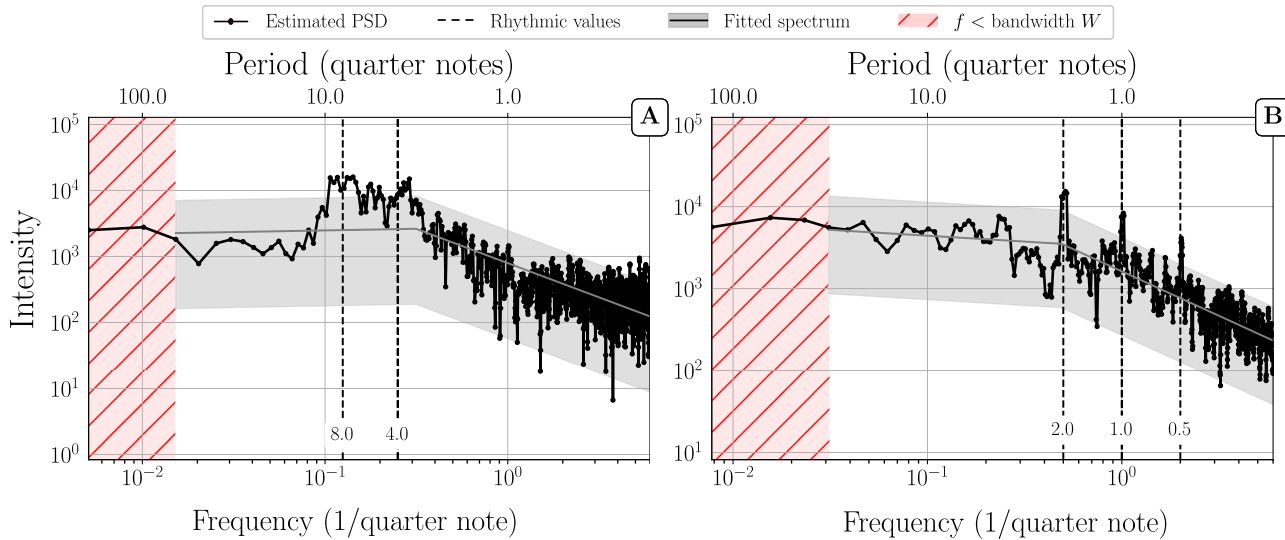

**Fig. 8 | Improvised solos.** Multitaper PSD estimation of (**A**) Benny Carter's solo on "I Got It Bad" and (**B**) Buck Clayton's solo on "Dickie's Dream". The shaded grey area represents a 95% confidence interval on the basis of the fitted PSD (grey line). Dashed vertical lines mark periods from 0.5 to 8 quarter notes. **A** shows that the rhythmic variability in some solos can considerably broaden the rhythmic peaks at 4 to 8 quarter notes. The flattening of the PSD at high frequencies is a quantization artifact: the solo contains rhythmic values shorter than the unit duration (1/12 quarter note) which results in a nearly random signal at the highest frequencies. **B** shows an example of a typical PL+P structure where rhythmic peaks are present. The peaks are visible but are not very pronounced compared to the peaks observed for classical music scores.

tradeoff between bias and variance allowed us to extend the PSD-estimates to the smallest possible frequencies and thus to show that the power laws of refs. 12–16 do not extend to arbitrarily small frequencies, but are crossing over to the plateaus (for musical pieces that are long enough). In a careful DFA analysis of a large corpus of musical compositions, Gonzalez-Espinoza et al.[16] moreover reported deviations from single power laws in the DFA fluctuation function in the form of more complicated profiles. Comparing the frequency range, where the deviations occur, it is very likely that these profiles are associated with the presence of periodic components in the musical time series, which can easily be resolved as rhythmic peaks in the PSD and appear as broad shoulders in the DFA (see e.g. supplementary information, Fig. 5). The PSD has the advantage of resolving the rhythmic peaks in this frequency range and can also detect the transitions to plateaus at small frequencies more conveniently than DFA.

Having summarized further results already at the end of our introduction, we here want to discuss the implications of the power-law + plateau (PL+P) structures. These structures represent strong and slowly decaying correlations that last up to a characteristic cutoff time, beyond which they quickly become insignificantly small. While slowly decaying correlations reflecting persistence seem to be a necessity without which music would appear too random, their finite-time horizon constitutes an additional element. We think that this element, which introduces a stronger degree of innovation and surprise than the pure power-law behavior, is substantial in order to maintain the interest and attention of listeners. We find cutoff times reaching from the typical lengths of phrases to the typical lengths of sections in musical form (A, B, C, ...) and thus they might reflect the innovation brought forth by new musical phrases or by musical form. At least and at last, new movements in symphonies and sonatas, which often introduce entirely new musical ideas and musical material, should break the persistence of slowly decaying correlations. While these cutoff times are piece-dependent and composer-dependent, a clear trend emerged from our results. Longer pieces were systematically associated with the appearance of plateaus, and when pure power-laws occurred occasionally without plateaus, this was predominantly in shorter pieces. This is a strong indication that even in shorter pieces

one may count with plateaus, that are merely masked due to the broader range in which the estimation is biased and would show up, if the PSD-bandwidth could be lowered.

In rare cases, the identification of a plateau turned out to be ambiguous. In these PL+P cases, the slope of the plateau (in the log-log-representation) showed a slight variability around 0 (see e.g. Andante of String Quartet No.18 in A major, K. 464" in supplementary Fig. 12). These variations might be the result of a large variance, or mark the beginning of a transition to a plateau in pieces that are too short to allow the PSD-estimation to reach lower frequencies. The behavior of the autocorrelation functions in such cases, however, is not substantially influenced by these small slopes and thus they have no influence on the conclusions of our work.

The finite range of slowly decaying correlations rules out the possibility of statistical self-similarity and never-ending correlations, that was evoked by the lack of bounds in the observation of $1/f$-noise by Voss and Clarke[12]. The findings shown in the present article for pitch time series are based on the analysis of many pieces by various composers from different styles and eras and emphasize the fact that musical time series decorrelate after a certain time. The fact that we did not identify any self-similar compositions does not imply that they are impossible but shows that statistical self-similarity is not a usual feature of musical compositions. While long-range correlated sequences of pitches can easily be generated algorithmically, creating them is usually not a goal for human composers. It would be worthwhile to search for self-similar compositions and to study how they might differ from normal musical pieces.

Power-law behavior also shows up in the PSDs of small random timing fluctuations in musical performances[21–24]. In contrast to the results reported in the present article, however, crossovers to plateaus and cutoff frequencies have not been reported in these cases. This important difference is probably due to the purely psychophysical nature of microtiming fluctuations. Similarly one finds power law behavior in sensorimotor experiments on tapping strength fluctuations[25]. Small random microtiming fluctuations are involuntary, they are ubiquitous in performances, and they do not play a musical role, as was shown e.g. for the swing feel in jazz[20]. This is not true for systematic microtiming deviations, however. They do have a function

in musical performances and they were shown to play a key role for the swing feel in jazz[19].

## Methods

### Data corpuses and data format

For ease of access and readability, we based our study on musical pieces stored in MIDI (".mid") format. It is commonly used in music production, performance and analysis. In this format, notes are reduced to their bare minimum in a representation that stores the pitch, the onset and the end of a note as well as the loudness and relative position of the note w.r.t the preceding note. MIDI files are therefore easy to interpret and can be processed rapidly by a computer.

The entirety of the data that we used in the present study stems from free collaborative online databases. For written compositions of classical music, we primarily used MIDI files from kunstderfuge.com[26], and completed our corpus with midiworld.com[27] and imslp.org[28] when needed. The MIDI files of improvised music were extracted from the database of the Weimar Jazz Database of the Jazzomat research project[29]. For our corpus of improvised pieces, we used every entry of this database, apart from pieces that were too short (less than 500 data points) and files that did not follow the standard MIDI format. Since we need one-dimensional time-series for our analysis, we prioritized files corresponding to monophonic instruments. When this was not possible, we restricted ourselves to the top voice (e.g., if two notes are played simultaneously, we consider only the top one). Thus, our study did not include pieces written for instruments like piano or harpsichord. For pieces where several instruments simultaneously play different melodic lines in a polyphonic manner we selected the one that stands out most (e.g. the first violin).

The complete list of pieces constituting the corpus of classical music scores and jazz improvisations we studied is given in the supplementary information.

### Power-spectral density

There are two prominent methods that can be used to characterize fluctuations in stochastic time series, power spectral density (PSD) analysis and detrended fluctuation analysis (DFA). While DFA has the advantage of being able to deal with multivariate time series, it is not well adapted to signals containing periodic components, which are reflected as broad shoulders in the DFA fluctuation function. This is discussed in more detail in the supplementary information. We found this side effect disadvantageous for our purpose, as many of our time series exhibit periodic components. In the PSD, on the other hand, periodic components show up as more or less pronounced peaks, which can easily be identified. We therefore decided to use PSD-analysis rather than DFA.

The power spectral density (PSD) of a signal $x(t)$, is defined as

$$P_{xx}(f) = |\hat{x}(f)|^2, \tag{1}$$

where $\hat{x}(f)$ is the Fourier transform of $x(t)$. It is closely related to the correlations present in the signal and can therefore be used to quantify them. In fact, under the assumption that the process studied is wide-sense stationary, the Wiener-Khinchin theorem states that the auto-correlation function $R_{xx}(t)$, is equal to

$$R_{xx}(t) = \int_{-\infty}^{\infty} P_{xx}(f) e^{ift} dt \tag{2}$$

PSDs following an inverse power-law satisfy the relation $P_{xx}(f) \propto f^{-\beta}$ where the exponent $\beta$ plays a special role. They are particularly relevant in long-range correlated processes. Indeed, if the PSD follows a pure inverse power-law, so does the auto-correlation function[30]. For a finite signal captured as a time-series, the PSD can only be estimated, however. A wide variety of methods have been developed to this

end[31,32]. We chose to conduct our analysis with non-parametric methods as the statistical process underlying music composition cannot be quantified. Such methods are usually based on windowing: the data are separated into several segments, in which the discrete Fourier transform is computed. The estimate is the average of the squared-modulus of each transformed segment. In other words, if we consider a discrete time-series $X(t)_{t \in 0,1,...T}$ truncated into M segments, and $X_n(t)$ is the $n$th segment of length L, the estimate $\hat{P}_{xx}(f)$ of the PSD is

$$\hat{P}_{xx}(f) = \left| \frac{1}{M} \sum_{n=1}^{M} \sum_{t=0}^{L-1} X_n(t) e^{-i(2\pi/L)ft} \right|^2. \tag{3}$$

As a result of windowing, the accessible frequency range of the results is limited, since segments are shorter than the original signal. To cope with this limitation, methods like Nutall-Carter or multitaper methods have been developed. In the present paper, we chose to use the multitaper method and will describe it below.

### Time series extraction

To study the power spectral density of musical pitch, we extracted pitch time-series from the MIDI files of our corpus. Given a MIDI file of a piece, the first step was to extract two sequences of integers corresponding to the pitch and durations of notes, respectively. This was done using the MIDI.jl and MusicManipulations.jl Julia packages[33], because they allow extraction of the durations in absolute MIDI time. The sequences of pitches follow the MIDI pitch convention and therefore range from 0 to 127. Time is measured in ticks, the standard time unit of MIDI format. To preserve the information associated to the temporal duration of the notes in the time-series, each note in the sequence was quantized and segmented according to a grid of unit duration of 1/12th of a quarter note (for more details see an example in Supplementary Fig. 1). This choice allows us to treat the two most prevalent time signatures (number of quarter notes per bar), 3/4 and 4/4, in a common framework for jazz and classical music. A value of 0 was used to indicate the absence of notes (see supplementary information for a detailed explanation).

This procedure keeps the onset and offset timings of notes and does not influence the asymptotic power-law behaviors for $f \to 0$ (or $t \to \infty$). It was already used in refs. 34 and 16; a difference of our approach is that we fixed the time unit of the grid to a piece-independent note value. This makes the process simpler to automatize and ensures that all our results are given in the same units. As a result of fixing the unit duration to 1/12th of a quarter note, the time period T corresponding to a given frequency f is obtained via

$$T = 1/12f. \tag{4}$$

To ease readability, the figures shown in this article include an additional horizontal axis showing the time periods corresponding to the bottom frequency axis.

**Multitaper method.** To estimate PSDs, we used the multitaper method[17]. As mentioned above, it is based on windowing and averaging. It has several advantages, compared to most non-parametric approaches: it does not limit the accessible frequency range of the results as much as other methods, it strongly reduces the amount of broadband spectral leakage and it allows for precise control of the local bias[35]. This is due to the type of windows (called tapers) used in multitaper estimation. They belong to the family of Slepian sequences and have desirable mathematical properties: the tapers of a given sequence are orthogonal. Thus, multiplying a time-series with tapers ensures statistical independence between the products, which reduces the variance upon averaging. Moreover, Slepian sequences are

solutions of the so-called spectral concentration problem[36], they effectively minimize the above mentioned broadband bias. The most relevant parameter of the multitaper method is the time-bandwidth product NW, where N is the length of the time-series. The parameter W is the *bandwidth*, which characterizes the spread of the tapers in frequency space[37]. The value of NW dictates the number of usable tapers K ($K < 2NW$) and governs the *bias-variance* trade-off. This trade-off will be detailed in the next subsection. With the multitaper method, the estimate $\hat{P}_{xx}$ is obtained as

$$\hat{P}_{xx}(f) = \left| \frac{1}{K} \sum_{k=1}^{K} \sum_{t=1}^{N} T_k(t) X(t) e^{-i(2\pi/N)ft} \right|^2, \quad (5)$$

where $T_k(t)$ is the $k^{th}$ taper of the sequence. Since the length $N$ is here the total length of the time-series, it makes the total frequency range of the results accessible. The data need to be centered around the mean before applying Eq 5 in order to eliminate potential spurious low-frequency components. For the estimations, we used the R package *spec.mtm*.

**Fitting procedure.** In order to identify potential trends in the PSDs more accurately (e.g. power-law decay), we apply a fitting to each estimate. Several piecewise linear fits of the PSD in log-log representation are optimized via a weighted least-square procedure. The weighting is necessary, since otherwise the log-log representation would give a bigger importance to high frequencies in the fit due to a higher density of points. A least-square based error function is used to select the best fit. Finally, we control for rarely occurring misfits and adapt the boundaries of fitting parameters, if necessary. The fitting parameter optimization was done using Python's lmfit package.

**Variance and bias.** In order to avoid misinterpretations in PSD estimations, we have to consider the potential sources of artifacts. It is therefore worthwhile to consider the bias and variance of PSD estimations in detail. The bias and variance of an estimation $\hat{P}_{xx}(f)$ are defined as

$$B(f) = P_{xx}(f) - \mathbb{E}\left[ \hat{P}_{xx}(f) \right] \quad (6)$$

$$\sigma^2(f) = \mathbb{E}\left[ (\hat{P}_{xx}(f) - \mathbb{E}[\hat{P}_{xx}(f)])^2 \right] \quad (7)$$

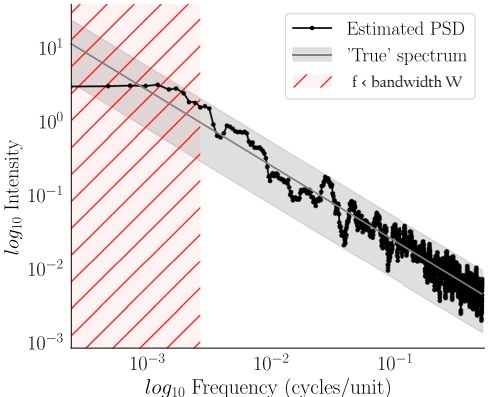

**Fig. 9 | Flattening bias.** The power spectral density of a simulated 1/*f*-noise time-series was estimated by the multitaper method with NW=2. In log-log representation the exact PSD follows a linear decay. The local bias, however, flattens the peak at the low end of the spectrum below the bandwidth frequency W. The grey shaded area represents a 95% confidence interval for the estimation based on the assumption of a linear relationship.

respectively, where $\mathbb{E}[.]$ denotes the expectation value. They measure the systematic and variance of $\hat{P}_{xx}(f)$ w.r.t. the true value of $P_{xx}(f)$. The multitaper method provides asymptotically unbiased and consistent estimations, which means that bias and variance can become arbitrarily small for time-series long enough[17,38]. However, the time-series we are dealing with are finite ($10^3 \sim 10^4$ data points), and thus taking these possible sources of error into account is essential for a correct interpretation of results. Let us consider the bias in more detail. In the case of multitaper, the *time-bandwidth* product NW determines the number of tapers K that should be used (typically, $K = 2NW - 1$). This effectively controls the magnitude of bias and variance. The higher NW, the more tapers are available, which implies more averaging and results in stronger variance reduction. On the other hand, a large value of NW can also mean a large bandwidth, which implies stronger bias. For the multitaper method, the bias is given by

$$B(f) = P_{xx}(f) - \frac{1}{K} \int_{-\pi}^{\pi} \sum_{k=1}^{K} |T_k(f-\nu)|^2 P_{xx}(f) d\nu \quad (8)$$

In frequency space, the sum of Slepian tapers is composed of a mostly flat region in the interval $[-W; +W]$ delimited by the bandwidth and a residual tail for frequencies outside this interval. The expression of Eq (8) can therefore be separated into a *local bias* $B_W(f)$

$$B_W(f) = P_{xx}(f) - \frac{1}{K} \int_{-W}^{W} \sum_{k=1}^{K} |T_k(f-\nu)|^2 P_{xx}(f) d\nu \quad (9)$$

representing the influence of the Slepian tapers on the power-spectrum inside the restricted region $[-W; W]$, and a *broadband bias* $B_B(f)$

$$B_B(f) = P_{xx}(f) - \frac{1}{K} \int_{\mathcal{D}} \sum_{k=1}^{k=K} |T_k(f-\nu)|^2 P_{xx}(f) d\nu \quad (10)$$

where $\mathcal{D} = [-\pi, \pi) \setminus (-W, W)$ represents the frequency region influenced by the above mentioned residual tail. The contribution of the broadband term, $B_B(f)$, was shown to be bounded and much smaller than its counterpart $B_W(f)$[17,39]. We will therefore only concern ourselves with the local bias term. As is apparent from Eq (9), it is the result of a convolution which mixes the values of $P_{xx}(f)$ at frequency scales smaller than the bandwidth W. This has two effects: it reduces the resolution, i.e. the ability to distinguish neighbouring peaks, and it flattens out peaks and troughs. We illustrate the effects of the local bias in Fig. 9, where we plot the estimated PSD of simulated pink noise. The bias becomes apparent as the peak at zero frequency is flattened in the frequency range of the bandwidth, thus creating a plateau as an artifact. Note that real plateaus are not affected by the bias.

Besides the bias, we also need an estimate for the variance of the PSD estimation. The variance is governed by the number of tapers used over which the averaging is performed. Under certain loose conditions[40,41] fluctuations around the true value of the PSD can be shown to follow:

$$\frac{2K\hat{P}_{xx}(f)}{P_{xx}(f)} \simeq \hat{\chi}^2_{2K} \quad (11)$$

where $\hat{\chi}^2_{2K}$ is the chi-square probability density distribution with 2K degrees of freedom, and K is the number of tapers used. This straightforwardly provides a confidence interval for our estimations. For example, given an estimate $\hat{P}_{xx}$, a 95% confidence interval for the true spectrum $P_{xx}(f)$ is:

$$\frac{2K\hat{P}_{xx}(f)}{\chi^2_{2K}(0.025)} < P_{xx}(f) < \frac{2K\hat{P}_{xx}(f)}{\chi^2_{2K}(0.975)} \quad (12)$$

**Reporting summary**

Further information on research design is available in the Nature Portfolio Reporting Summary linked to this article.

## Data availability

The data used in this study was obtained from free collaborative online databases that are openly accessible (see subsection Data corpuses and data format). The complete list of pieces constituting the corpus of classical music scores and jazz improvisations we studied is given in the supplementary information. To ease the reproducibility of our results, we also provide the data used in the present article in a figshare repository[42].

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

## Acknowledgements

We acknowledge generous support of the Max Planck Society in the form of an emeritus group.

## Author contributions

T.G.: Conceived the study. C.N., and T.G.: Performed research. C.N.: Formatted and analyzed the data. C.N.: and T.G.: Wrote the article.

## Funding

## Competing interests

The authors declare no competing interests.
