## [Peer Review File · Nature Communications]

REVIEWER COMMENTS

Reviewer #1 (Remarks to the Author):

The authors of "Stochastic Properties of Musical Timeseries" perform a power spectrum analysis of classical music pieces as well as improvised performances by jazz musicians, looking for long-range autocorrelations. A multitaper method is used, which allows an improved estimation of the power spectrum on very slow frequency components corresponding to long time scales. Using a piecewise fitting procedure that adjusts power laws of type $f^{(-\beta)}$, they extracted frequency ranges with $\beta > 0$, indicating linear autocorrelations, and frequency bands with $\beta = 0$ that are located at the edge of the slow frequencies. The transition between the two regimes marks the time scale at which autocorrelation vanishes. The authors claim that the time scale of vanishing autocorrelations can serve as a measure of the degree of consistency and predictability in music and could be characteristic of composers. While the observation of time scales with abruptly disappearing autocorrelations in otherwise strongly correlated time series is an interesting observation, I have several major concerns about this contribution.

1) When looking for long range autocorrelations it is advisable to use Detrended Fluctuation Analysis (DFA), which could additionally account for possible nonstationarities of the time series. I would strongly recommend that the authors confirm their results using DFA. They should check whether the adjustment of power laws (or straight line in a log-log-representation) is justified. Some of the log-log plots of the empirical data show strong oscillatory behavior, which makes the adjustment of straight lines at least problematic. In addition, DFA methods can also be applied to multivariate time series, so that the restriction to the "outstanding instrument" or the "top voice" would no longer be necessary.

2) In the introduction the authors refer to several former contributions of a similar analysis presenting quite contradicting results (references 12 to 16). How do the authors relate their results to these previous findings, where β -values vary between 0.5 to 2. In particular, in reference [16] it was reported that the profiles of the DFA fluctuation function (corresponding to the power spectral density estimate) do not follow a power law but describe more complicated profiles. Although the authors used partly the same data pool than in [16], it seems that they adjusted always straight lines. How do they explain this discrepancy?

3) In the second and third line of page three the authors claim that red noise or signals with $\beta = 2$ are associated with fast exponential decay of linear autocorrelations. This statement is simply wrong. Signals with $\beta = 2$, like the one-dimensional random walk, show strong long-range autocorrelations and the autocorrelation function by no means decays like an exponential.

4) The authors should be careful when using the term noise. While it is true that noise signals may be strongly autocorrelated over extremely large time scales, not any autocorrelated signal with some β -value is noise. Noise signals are characterized by Fourier phases, which are independent random numbers uniformly distributed between zero and 2π , with an arbitrary distribution of Fourier amplitudes (power spectral density). The Wiener-Khinchin theorem states the equivalence between the linear autocorrelation function and the power spectral density. Hence, other properties like nonlinear correlations or determinism are somehow encoded in spectra of the Fourier phases. Note, the whole theory of Fourier transform surrogates is based on this fact.

5) The authors state at page three “The transition from a strongly correlated to an uncorrelated time series at large time differences reflects the interplay between predictability and uncertainty, between expectation and surprise in music as discussed by music theorists”. However, a balance between predictability and surprise can also be assigned to signals with power law characteristics with $\beta=1$, while stronger autocorrelations ($\beta>1$) are associated with an excess of predictability or too monotonous behavior and weakly autocorrelated signals ($\beta<1$) could be perceived as too disordered, i.e. noise-like. I cannot see why an abrupt transition to zero autocorrelations at a well-defined time scale is necessary to achieve such a balance. I feel that the above-mentioned statement of the authors is just an unproven claim. Please comment on that.

6) The cutoff time varies over two orders of magnitude (from 4 to 100 quarter note units). How is it possible that a parameter such as the cutoff time scale, which is supposed to define a finely tuned balance between predictability and surprise, varies so much? If so, it seems that this balance is not a universal quantity.

7) Is there any correlation between the cutoff time of a musical piece and the β -value of the scaling of its power spectral density?

8) The authors claim (final part of page 3) that the cutoff time of the transition between strong to zero autocorrelation can be used to characterize different composers, stating that their findings indicate that the cutoff time estimated for Mozart are typically larger than in Bach's. They refer in particular to Fig. 3 and 4 to substantiate this statement. However, these figures only show individual examples of a single piece of each composer that cannot indicate a trend, i.e. the authors do not provide any evidence for their statement. Furthermore, if this is by trend true for the pieces they analyzed, it is not a prove for such a strong general statement referring to any composer.

9) The authors state in the second half of page 10 “The appearance of a plateau, is strongly related to the total length of the piece, as only shorter pieces tend to present PL shapes”. The length of the

cutoff periods is therefore not meaningful for composers, i.e. they only apply to special cases in which composers create exclusively long or short pieces, which is certainly not what the authors intended. The authors therefore contradict themselves in the interpretation of their results.

10) Note, intuitively it is understandable that the cutoff times are directly correlated to the length of a musical piece. A longer theme requires more time to work through, i.e. a piece of music that contains longer themes also has a longer total length. The length is also determined by what you can think of for a theme, how you vary it, modify it etc. Hence, if the cutoff time is directly related to the length of a theme or musical motif the correlation between cutoff length and the length of a musical piece is a trivial issue. This could easily be checked quantitatively.

11) In Eq. (8) the integral runs from $-1/2$ to $+1/2$. How does that come about? Is it assumed that the time interval between successive samples is unity, such that ω runs between these borders and f between $\pm\pi$? In this case, the integration variable should be angular velocity and not frequency.

12) The authors claim that they used 99 scores from classical music and 454 from improvised jazz. However, if we add up the number of single and several movements as well as improvised solos in Table 1, we arrive at 101 and 450 respectively. Please comment on that.

13) Referring to Fig. 7 the authors claim that “the times at which the plateaus appear in improvised solos are systematically shorter than the cutoff times observed in the case of classical music”. On closer inspection of Fig. 7, however, it is noticeable that the pieces of jazz improvisation are almost all shorter than 1000 quarter notes, while the length range of the classical music pieces is greater than 4000 quarter notes. If one compares Fig. 7.a and b within the range of 1000 quarter notes, both distributions seem to be very similar, and I believe no significant difference could be detected. If that is true, the interpretation of the authors is not true. Also, the roughly linear dependence of the cutoff period on the total length of a musical piece seems comparable.

14) On page 17 the authors claim that “Overall, when rhythmic peaks are visible in PSDs of jazz solos, they tend to be less pronounced than in classical compositions. This can be seen e.g. in Fig. 10A”. Again, in Fig. 10A a single case is shown, and no general trend can be derived from this result. Furthermore, inspecting Fig. 10B one gains the impression that these peaks are very sharp, at least for the jazz solo presented in this figure. Quantitative evidence in the statistical sense should be provided when general statements are made.

15) Several results in Fig. 5 of the supplementary material indicate that the plateau is not characterized by a slope with $\beta=0$, but it encounters negative values indicating anticorrelation on

large length scales. For me, this is a surprising result, but it is not mentioned anywhere in the manuscript. Could you comment on that?

Summarizing these observations, it appears that the plateau observed by the authors is not related to some kind of balance between predictability and surprise, nor does it serve as a characteristic biomarker of composers, but merely correlates with the overall length of a piece of music. Longer pieces of music offer more space for longer motifs and themes, which in turn form correlated units. This provides a trivial but consistent explanation for the observation of a plateau on large length scales. This phenomenon obviously has nothing to do with the fine-tuning of structured and disordered time scales.

Minor comments:

- 1) In the introduction the authors mention that with a special interest in chaos and dynamical systems new techniques of times series analysis were developed, citing literature that refer to Detrended Fluctuation Analysis and Transfer Entropy. However, linear correlations as detected by power spectral densities or DFA are by no means related to dynamical systems or chaos and statistical interdependencies between times series as quantified by Transfer Entropy are not related to such fields. In this sense, their comment is quite misleading.
- 2) The authors used the notation Wiener-Khinchin and Wiener-Khintchine, sometimes written in capital letters, sometimes in lower case. I would recommend using exclusively Wiener-Khinchin.
- 3) Multitaper method is sometimes written with capital letters and sometimes in lower case.
- 4) There are several typos like e.g. “pronounced”.

Reviewer #2 (Remarks to the Author):

The authors conducted a study on the stochastic properties of Classical and Jazz musical time series through power spectral analysis. Employing a multitaper method, they aimed to extend their estimations to smaller frequencies, classifying their analysis into power-law (PL) and power law with plateau (PL with P). The presentation of their work is commendable, with thorough explanations in all sections. The Introduction provides a comprehensive review of relevant literature, and the Results and Discussion sections are well-articulated. The clarity of the Data and Method section is notable. However, I have a query regarding Figure 7. Have the authors considered plotting both panels in a log-log scale? This adjustment might reveal a more discernible pattern.

Reviewer #3 (Remarks to the Author):

Report on the paper „Stochastic Properties of Musical Timeseries“

The method

In the paper, pitch autocorrelation in musical time series is discussed based on music pieces available in MIDI format. Classical music pieces are compared with Jazz solos. In polyphonic pieces, the authors focus on that instrument which stands out most. Extracted time series are quantized and segmented. The power spectral density (PSD) is analyzed by means of a multitaper smoother. As a first general result, a flat region of the PSD, determined by the bandwidth, is identified for very small frequencies. To characterize the uncertainty, a confidence band is estimated for the multitaper smoother.

The results

The paper establishes noteworthy results which add important findings to the literature discussed in the Introduction of the paper.

Major comments

In most music pieces, smoothing identified a plateau of the PSD for small frequencies beyond the above mentioned flat region. This plateau corresponds to zero autocorrelations from a certain lag on.

- The authors argue that such a plateau is characteristic for musical time series (page 17 in the discussion). That it is not found in every music piece is accounted to the relatively large bandwidth in shorter pieces. Plateaus would show up if the bandwidth could be lowered (page 10, line 6 from below). In my opinion, the authors should try to confirm this conjecture by, e.g., using an alternative bandwidth generator. Would, e.g., $NW = 1.5$ be possible?

- Concerning the plateau, the supplement shows a specificity for (at least) two of the interpreters of Jazz solos (Art Pepper, Bob Berg). For these interpreters, instead of a plateau, a rising line is estimated. Is this an artefact, could this be interpreted?

- For the end of the plateau, the so-called “cutoff period”, Figure 7 shows an approximately linear relationship to the length N of the piece. Obviously, the slope of this relationship is different for classical and Jazz pieces (for Jazz pieces, you might want to restrict the linear relationship to the interval $N \in (0, 500)$). Could this also be utilized for the comparison of classical and Jazz pieces?

The other main result of the paper is the characterization of the decay of the PSD and the autocorrelations.

- Figure 8 shows histograms of the exponents of the PSD curves. For classical music, a distinguished peak can be identified in 1. For the Jazz solos, I would identify a broader peak area between -2 and -0.75 (as the grey line indicates). The comments on page 17 (right before the discussion) appear to me as a slight overinterpretation (esp. the conjecture of a bi-modal distribution).

Last, but not least, from the abstract I had expected more interpretation on autocorrelations. For a reader, it might also be interesting what the relation of the “cutoff lags” to the length of the pieces is, in addition to the behavior of the “cutoff periods”. However, autocorrelations are only mentioned once in the results part, namely in Fig 5 and in its brief discussion on page 10. In the final discussion of the paper, in contrast, correlations are more intensively used for summarizing the results (pp. 17/18).

Minor comments

- In general, the term “time series” should be used instead of “timeseries” and “time-series” which both appear in the paper.

- On page 7, line 5 from below, please replace “standard deviation” by “variance”.

- On page 9, the authors mention in a footnote that 3 tapers can be used. Please shift the footnote to the main text and add a brief justification.

- On page 10, in the paragraph beginning with “In order not to surcharge ...” and ending with “ ... Fig 4A)” the authors should make a clearer distinction between parts related to the supplement of the paper (e.g. the Figures) and those related to the main body (e.g. Table 1).

- On page 13, the authors use the term “longer cutoff periods”. I would prefer to say “larger cutoff periods/values” in consistency with the definition and later notation.

- On page 15, in the legend of Fig 8, replace “solos from from the WJD” by “solos from the WJD”.

PSD paper: reply to reviewers

We would like to thank the three reviewers very much for their careful reading, their comments and suggestions. Their suggestions were helpful; in some cases two reviewers made similar suggestions, in one case all three of them: Reconsidering and replotting Fig 7 in a log-log representation revealed new details.

We addressed all of the reviewers' comments and followed their advice wherever feasible, e.g. by adding 8 new figures and text in the supplementary information. Below please find a point-by-point response to the reviewers' comments.

Please note:

Our answers to the reviewers' comments are marked in blue.

Our changes in the manuscript and in the supplementary information are marked in green, apart from the figure captions.

Fig 7 of the manuscript was re-done in a log-log representation.

In the supplementary information there are 8 new figures (Figs 3-10).

References [Rn] refer to the references below this reply, while references [n] refer to the references of the manuscript.

Reviewer #1

1) When looking for long range autocorrelations it is advisable to use Detrended Fluctuation Analysis (DFA), which could additionally account for possible nonstationarities of the time series. I would strongly recommend that the authors confirm their results using DFA. They should check whether the adjustment of power laws (or straight line in a log-log-representation) is justified. Some of the log-log plots of the empirical data show strong oscillatory behavior, which makes the adjustment of straight lines at least problematic. In addition, DFA methods can also be applied to multivariate time series, so that the restriction to the "outstanding instrument" or the "top voice" would no longer be necessary.

We agree with the reviewer's statement of the advantages of the DFA method, in fact, it was one the first methods we started using at the beginning of this project, but we realized that it

was not appropriate for our purpose, as it is very sensitive to periodic components, and this can impact the results, see e.g. [R1], [R2].

Such effects can be seen in the figure below, which shows the impact of periodic components on the DFA fluctuation function of simulated pink noise. They can lead to broad shoulders, broad enough that they can interfere with the power law regime. The PSD on the other hand can detect and resolve the periodic peaks and is more appropriate and meaningful in this situation. Note that in carrying out the power-law fits of the PSD we did not include the oscillatory regimes.

A method to mitigate the periodicity-induced bias in DFA was developed in [R3], but it heavily relies on a **manual intervention** based on an interpretation of the power-spectrum (with some arbitrariness) as well as user chosen delay embedding parameters, which in turn also affect the final DFA results. Musical pieces often show salient periodicity on different levels, expressed in pitch or rhythmic motifs, as well as in the overall structure. For these reasons DFA appeared to be less suitable and less reliable for our purposes, and we decided to use power spectral analysis instead.

We added a paragraph to the manuscript under the materials and methods section (see the beginning of “power spectral density”) addressing the advantages and disadvantages of DFA and explaining why we decided not to use it. Moreover we included more details and the figure on the impact of periodic components in the supplementary information in a section on “detrended fluctuation analysis”.

2) In the introduction the authors refer to several former contributions of a similar analysis presenting quite contradicting results (references 12 to 16). How do the authors relate their results to these previous findings, where β -values vary between 0.5 to 2. In particular, in reference [16] it was reported that the profiles of the DFA fluctuation function (corresponding to the power spectral density estimate) do not follow a power law but describe more complicated profiles. Although the authors used partly the same data pool than in [16], it seems that they adjusted always straight lines. How do they explain this discrepancy?

In fact our results as well as those reported in [16] indicate, indeed, that the power laws do not show a single (universal) exponent, but rather a continuum of possible values. Insofar there is no discrepancy. Differences can be seen in the frequency range (or time window range) in which the rhythmic peaks occur. The differences have a natural explanation, however. As illustrated in the above figure, periodic components are represented as relatively sharp peaks in the PSD, while they are represented by broader shoulders in the DFA fluctuation function. So there is not really a contradiction in the high-frequency or small time window regimes; the more complicated profiles in DFA appear to be another expression of the existence of periodic components in the time series. We think it is more appropriate and meaningful for these regimes, however, to use the PSD, as it can explicitly uncover the rhythmic peaks. Moreover, since in the PSD the rhythmic peak regime can easily be identified and delimited, it was possible to delimit the power law regime and to adjust straight lines.

With the PSD it is also easier (and thus more appropriate) to detect transitions to uncorrelated behavior with increasing time, corresponding to transitions to a plateau in the PSD. In our work, we did our best to control the accuracy of the estimates (by controlling bandwidth, bias, and variance).

We added a paragraph to the discussion section of the manuscript (second paragraph), to compare our results with the previous literature.

3) In the second and third line of page three the authors claim that red noise or signals with $\beta=2$ are associated with fast exponential decay of linear autocorrelations. This statement is simply wrong. Signals with $\beta=2$, like the one-dimensional random walk, show strong long-range autocorrelations and the autocorrelation function by no means decays like an exponential.

This is true of course, and having worked on Lévy flights and walks we are aware of this fact. But this claim is not exactly what we wrote, however. In writing ... " $1/f^2$ -spectra ... **may** be associated with a fast exponential decay of correlations" (which is the case e.g. when the spectrum follows a Lorentzian) we intended to emphasize the diversity of possible

autocorrelation decays associated with the observation of different exponents, also in order to show that the previous literature has left important open questions.

In the manuscript, in order to avoid misunderstandings we now made this statement more explicit and more accurate. Thanks for pointing it out.

4) The authors should be careful when using the term noise. While it is true that noise signals may be strongly autocorrelated over extremely large time scales, not any autocorrelated signal with some β -value is noise. Noise signals are characterized by Fourier phases, which are independent random numbers uniformly distributed between zero and 2π , with an arbitrary distribution of Fourier amplitudes (power spectral density). The Wiener-Khinchin theorem states the equivalence between the linear autocorrelation function and the power spectral density. Hence, other properties like nonlinear correlations or determinism are somehow encoded in spectra of the Fourier phases. Note, the whole theory of Fourier transform surrogates is based on this fact.

Thanks for stressing this point. Unless when citing other authors, we formulated the corresponding passages more carefully, by replacing or complementing the word “noise” where misleading.

5) The authors state at page three “The transition from a strongly correlated to an uncorrelated time series at large time differences reflects the interplay between predictability and uncertainty, between expectation and surprise in music as discussed by music theorists”. However, a balance between predictability and surprise can also be assigned to signals with power law characteristics with $\beta=1$, while stronger autocorrelations ($\beta>1$) are associated with an excess of predictability or too monotonous behavior and weakly autocorrelated signals ($\beta<1$) could be perceived as too disordered, i.e. noise-like. I cannot see why an abrupt transition to zero autocorrelations at a well-defined time scale is necessary to achieve such a balance. I feel that the above-mentioned statement of the authors is just an unproven claim. Please comment on that.

We fully agree with this possibility, and it is true that an abrupt transition to zero autocorrelations is not a necessary condition to achieve such a balance between predictability and uncertainty. In fact, we had the option that this balance could be achieved by long-range correlations with $\beta=1$ among our expectations, before we started the project. But empirically we see that nature/humans/composers seem to have chosen a different option to realize the presence of predictability and uncertainty in musical time series.

In the manuscript we have slightly modified the corresponding sentence in order to avoid the misunderstanding of a necessary condition.

6) The cutoff time varies over two orders of magnitude (from 4 to 100 quarter note units). How is it possible that a parameter such as the cutoff time scale, which is supposed to define a finely tuned balance between predictability and surprise, varies so much? If so, it seems that this balance is not a universal quantity.

We think that 'balance' is not the optimal wording for this effect, as it might suggest a fine-tuning. The fact that the cutoff times vary between 4 to 100 quarter note units shows that the predictability varies among compositions (and as a trend to some extent even among composers), thereby reflecting structural choices. The cutoff time is not a universal quantity, but its mere existence in musical time series seems to be universal, as we found no sufficient indications for its absence.

In order to avoid the notion of fine-tuning, we avoided the word 'balance' and used the word 'interplay' instead.

7) Is there any correlation between the cutoff time of a musical piece and the β -value of the scaling of its power spectral density?

We checked this possibility. The following scatter plots show for the different data sets that there is no correlation between the two quantities.

We included this figure in a new subsection of the supplementary information.

8) The authors claim (final part of page 3) that the cutoff time of the transition between strong to zero autocorrelation can be used to characterize different composers, stating that their findings indicate that the cutoff time estimated for Mozart are typically larger than in Bach's. They refer in particular to Fig. 3 and 4 to substantiate this statement. However, these figures only show individual examples of a single piece of each composer that cannot indicate a trend, i.e. the authors do not provide any evidence for their statement. Furthermore, if this is by trend true for the pieces they analyzed, it is not a proof for such a strong general statement referring to any composer.

What we meant is that the cutoff time as a measure for the degree of persistence can characterize different compositions and - only in some cases but not generally - also different composers. We observed this by comparing Mozart's and Bach's compositions in the corpuses we studied, and mentioned it without giving further details in the introduction. So far a few more details were given on page 10, where we also pointed out that this is not without exceptions. This observation was not based on Figures 3 and 4. As this observation cannot be generalized to any two composers, we previously did not want to put too much emphasis on it and refrained from going into more details.

In order to avoid the misunderstanding that composers might generally be distinguished by different cutoff times, we now toned down this statement in the introduction.

Moreover, in order to give more details, we have now included a new figure, Fig. 7, in the supplementary information (see also below), which gives a detailed comparison of the cutoff times in all the analyzed movements by Bach and Mozart and clearly reveals the trend we mentioned. We now also make this more explicit in the main text on page 10 and refer to Fig. 7 of the supplementary information. We do not claim that this can be done for any two composers.

9) The authors state in the second half of page 10 “The appearance of a plateau, is strongly related to the total length of the piece, as only shorter pieces tend to present PL shapes”. The length of the cutoff periods is therefore not meaningful for composers, i.e. they only apply to special cases in which composers create exclusively long or short pieces, which is certainly not what the authors intended. The authors therefore contradict themselves in the interpretation of their results.

We agree, this is not what we wanted to imply, please see also our reply to 8) above. We merely wanted to state that by comparing certain composers it seems to be possible to observe a trend (not without exceptions) in the cutoff times of their compositions. We toned down the text in order to avoid misunderstandings, see 8).

10) Note, intuitively it is understandable that the cutoff times are directly correlated to the length of a musical piece. A longer theme requires more time to work through, i.e. a piece of music that contains longer themes also has a longer total length. The length is also determined by what you can think of for a theme, how you vary it, modify it etc. Hence, if the cutoff time is directly related to the length of a theme or musical motif the correlation between cutoff length and the length of a musical piece is a trivial issue. This could easily be checked quantitatively.

Investigating the correlations between cutoffs and theme/motif length in a systematic way is not straightforward as themes are usually composed of a variable number of subunits that are altered over the course of the piece. Determining *reliably* what constitutes the main theme as well as when it starts and ends by automatic musical structure and theme detection is quite difficult and is a research topic in itself, which lies outside the scope of the current work, see e.g. Maddage [R5] and Serra et al. [R6]. For this reason, we could not conduct an in depth analysis of the relationship between cutoff periods and the different lengths (sub-motif, motif, theme) at play within musical pieces. What we did, however, was to check for the effect of repetition at the level of individual movements. We plotted the cutoff lengths of several pieces from different composers as a function of the mean movement length in the corresponding piece. We see that there is no correlation between these two variables, which suggests that a profound investigation of the relationship between motif and theme lengths and cutoff times would probably not reveal any correlation. See the figure below; for more details please refer to the supplementary information.

We now included this figure and accompanying text with details in the supplementary information.

11) In Eq. (8) the integral runs from $-1/2$ to $+1/2$. How does that come about? Is it assumed that the time interval between successive samples is unity, such that ω runs between these borders and f between $\pm\pi$? In this case, the integration variable should be angular velocity and not frequency.

This is correct. We changed the integration boundaries to $\pm\pi$ in order to make it consistent with the rest of the manuscript. Thank you for pointing that out.

12) The authors claim that they used 99 scores from classical music and 454 from improvised jazz. However, if we add up the number of single and several movements as well as improvised solos in Table 1, we arrive at 101 and 450 respectively. Please comment on that.

This project ran over a long period of time, and as more and more pieces got added to the dataset, we made a counting mistake. We are grateful for the correction.

13) Referring to Fig. 7 the authors claim that “the times at which the plateaus appear in improvised solos are systematically shorter than the cutoff times observed in the case of classical music”. On closer inspection of Fig. 7, however, it is noticeable that the pieces of jazz improvisation are almost all shorter than 1000 quarter notes, while the length range of the classical music pieces is greater than 4000 quarter notes. If one compares Fig. 7.a and b within the range of 1000 quarter notes, both distributions seem to be very similar, and I believe no significant difference could be detected. If that is true, the interpretation of the authors is not true. Also, the roughly linear dependence of the cutoff period on the total length of a musical piece seems comparable.

Thank you for this remark. Similarly, reviewer 2 suggested to plot Fig. 7 in a log-log representation. This gave some new insight, indeed, see figure below. There seems to be a stronger trend for classical pieces than for jazz improvisations. This representation also reveals a vertical accumulation of data points around 128 and 256 quarter notes, which has a simple explanation (see main text).

We replaced Fig. 7 by a log-log representation, adapted the text accordingly, and toned down the statement pertaining to trends.

14) On page 17 the authors claim that “Overall, when rhythmic peaks are visible in PSDs of jazz solos, they tend to be less pronounced than in classical compositions. This can be seen e.g. in Fig. 10A”. Again, in Fig. 10A a single case is shown, and no general trend can be derived from this result. Furthermore, inspecting Fig. 10B one gains the impression that these peaks are very sharp, at least for the jazz solo presented in this figure. Quantitative evidence in the statistical sense should be provided when general statements are made.

This is true, the figure shows only two examples, we gave many more examples in the figures of the supplementary information. A histogram showing the distribution of peak-to-baseline in the case of classical compositions vs. jazz improvisations is now included with accompanying text in the supplementary information to underpin this statement *quantitatively* (see also figure below). Overall, we see that on average, rhythmic peaks tend to be much more pronounced in classical compositions than in jazz improvisations.

15) Several results in Fig. 5 of the supplementary material indicate that the plateau is not characterized by a slope with $\beta=0$, but it encounters negative values indicating anticorrelation on large length scales. For me, this is a surprising result, but it is not mentioned anywhere in the manuscript. Could you comment on that?

Cases with negative slopes at low frequencies are usually linked to large fluctuations in that range causing the fitting routine to identify slopes where there might be none. Nevertheless, in some rare examples one might argue that these trends are not resulting from the fitting procedure and are indeed part of the musical time series.

To account for such cases, and ensure the validity of our results, we ran several noise simulations allowing for a slight (positive and negative) slope in the plateau part of the corresponding power spectrum, for more details see a new subsection of the supplementary information and the figure below. We found that such slight slopes did not have an appreciable effect on the autocorrelation function. We therefore did not want to infer any anticorrelations.

Summarizing these observations, it appears that the plateau observed by the authors is not related to some kind of balance between predictability and surprise, nor does it serve as a characteristic biomarker of composers, but merely correlates with the overall length of a piece of music. Longer pieces of music offer more space for longer motifs and themes, which in turn form correlated units. This provides a trivial but consistent explanation for the observation of a plateau on large length scales. This phenomenon obviously has nothing to do with the fine-tuning of structured and disordered time scales.

This is a summary. We answered these points in detail already above, in particular under 10) and 8).

Minor comments:

1) In the introduction the authors mention that with a special interest in chaos and dynamical systems new techniques of times series analysis were developed, citing literature that refer to Detrended Fluctuation Analysis and Transfer Entropy. However, linear

correlations as detected by power spectral densities or DFA are by no means related to dynamical systems or chaos and statistical interdependencies between times series as quantified by Transfer Entropy are not related to such fields. In this sense, their comment is quite misleading.

This was meant in a historical, not a factual context. We modified the text.

2) The authors used the notation Wiener-Khinchin and Wiener-Khintchine, sometimes written in capital letters, sometimes in lower case. I would recommend using exclusively Wiener-Khinchin.

done

3) Multitaper method is sometimes written with capital letters and sometimes in lower case.

done

4) There are several typos like e.g. "pronouced".

done

Reviewer #2:

The authors conducted a study on the stochastic properties of Classical and Jazz musical time series through power spectral analysis. Employing a multitaper method, they aimed to extend their estimations to smaller frequencies, classifying their analysis into power-law (PL) and power law with plateau (PL with P). The presentation of their work is commendable, with thorough explanations in all sections. The Introduction provides a comprehensive review of relevant literature, and the Results and Discussion sections are well-articulated. The clarity of the Data and Method section is notable. However, I have a query regarding Figure 7. Have the authors considered plotting both panels in a log-log scale? This adjustment might reveal a more discernible pattern.

This is a very good suggestion, and we thank the reviewer for that. Plotting Figure 7 in a log-log scale made the differences between both panels more salient and revealed new details. We therefore replaced the old version of Figure 7 by a log-log representation. For more details see also our reply to comment 13) of reviewer 1 above.

Reviewer #3:

Report on the paper „Stochastic Properties of Musical Timeseries“

The method

In the paper, pitch autocorrelation in musical time series is discussed based on music pieces available in MIDI format. Classical music pieces are compared with Jazz solos. In polyphonic pieces, the authors focus on that instrument which stands out most. Extracted time series are quantized and segmented. The power spectral density (PSD) is analyzed by means of a multitaper smoother. As a first general result, a flat region of the PSD, determined by the bandwidth, is identified for very small frequencies. To characterize the uncertainty, a confidence band is estimated for the multitaper smoother.

The results

The paper establishes noteworthy results which add important findings to the literature discussed in the Introduction of the paper.

Major comments

In most music pieces, smoothing identified a plateau of the PSD for small frequencies beyond the above mentioned flat region. This plateau corresponds to zero autocorrelations from a certain lag on.

- The authors argue that such a plateau is characteristic for musical time series (page 17 in the discussion). That it is not found in every music piece is accounted to the relatively large bandwidth in shorter pieces. Plateaus would show up if the bandwidth could be lowered (page 10, line 6 from below). In my opinion, the authors should try to confirm this conjecture by, e.g., using an alternative bandwidth generator. Would, e.g., $NW = 1.5$ be possible?

This is a good suggestion, as it may illustrate the role played by the bandwidth in uncovering or concealing a plateau, but it does not come without some problems. In the first example below one can see that for $NW = 2$ the plateau is not detected by the fitting algorithm and one might debate whether or not it is present, whereas in the case of $NW = 1$, a plateau shows up. On the other hand, lowering NW below 2 considerably enhances the variance and makes conclusions less reliable.

PSD estimation of David Liebman's Begin the Beguine with two different NW products: NW = 2 (left) and NW = 1 (right). Lowering the NW products reduces the bandwidth of the bias and uncovers a plateau, but considerably increases the variance.

In order to illustrate the effect of the bandwidth we therefore included the above and the following figure in the supplementary information. It compares two situations with sufficiently small variance for NW = 12 and NW = 3 and shows how an existing plateau can be concealed depending on the variance.

PSD estimation of Beethoven's 4th Symphony, Opus 60, 2nd movement obtained using two different NW products: 12 (left) and 3 (right). A high NW product can lead to a bandwidth so large that it conceals the plateau appearing at around period 80.

- Concerning the plateau, the supplement shows a specificity for (at least) two of the interpreters of Jazz solos (Art Pepper, Bob Berg). For these interpreters, instead of a plateau, a rising line is estimated. Is this an artefact, could this be interpreted?

This point was also raised by reviewer 1. We responded to it under comment 15) and included a new subsection and a figure in the supplementary information.

- For the end of the plateau, the so-called “cutoff period”, Figure 7 shows an approximately linear relationship to the length N of the piece. Obviously, the slope of this relationship is different for classical and Jazz pieces (for Jazz pieces, you might want to restrict the linear relationship to the interval $N \in (0, 500)$). Could this also be utilized for the comparison of classical and Jazz pieces?

Similar points were raised by reviewers 1 and 2. Reviewer 2 suggested using a log-log representation for the figure, which turned out to be quite useful and revealed new details. Please see under comment 13 of reviewer 1 for more details. We included the new figure and in the text discuss the different slopes for classical and for jazz pieces (merely different trends).

The other main result of the paper is the characterization of the decay of the PSD and the autocorrelations.

- Figure 8 shows histograms of the exponents of the PSD curves. For classical music, a distinguished peak can be identified in 1. For the Jazz solos, I would identify a broader peak area between -2 and -0.75 (as the grey line indicates). The comments on page 17 (right before the discussion) appear to me as a slight overinterpretation (esp. the conjecture of a bi-modal distribution).

We followed your advice, eliminated the conjecture of a bi-modal distribution, and mention the “broader peak” in the histogram for jazz solos.

Last, but not least, from the abstract I had expected more interpretation on autocorrelations. For a reader, it might also be interesting what the relation of the “cutoff lags” to the length of the pieces is, in addition to the behavior of the “cutoff periods”. However, autocorrelations are only mentioned once in the results part, namely in Fig 5 and in its brief discussion on page 10. In the final discussion of the paper, in contrast, correlations are more intensively used for summarizing the results (pp. 17/18).

Thanks for pointing out that it is worthwhile mentioning the implications for the autocorrelation function in more detail in the main body of the text than was done before. We followed your advice and now describe these implications more explicitly in the text in several places of the results section.

Minor comments

- In general, the term “time series” should be used instead of “timeseries” and “time-series” which both appear in the paper.

Done

- On page 7, line 5 from below, please replace “standard deviation” by “variance”.

Done

- On page 9, the authors mention in a footnote that 3 tapers can be used. Please shift the footnote to the main text and add a brief justification.

Done

- On page 10, in the paragraph beginning with “In order not to surcharge ...” and ending with “ ... Fig 4A)” the authors should make a clearer distinction between parts related to the supplement of the paper (e.g. the Figures) and those related to the main body (e.g. Table 1).

Done

- On page 13, the authors use the term “longer cutoff periods”. I would prefer to say “larger cutoff periods/values” in consistency with the definition and later notation.

Done.

- On page 15, in the legend of Fig 8, replace “solos from from the WJD” by “solos from the WJD”.

Done.

References

[R1] Hu, K., Ivanov, P. C., Chen, Z., Carpena, P., & Stanley, H. E. (2001). Effect of trends on detrended fluctuation analysis. *Physical Review E*, 64(1), 011114.

[R2] Marković, D., and M. Koch (2005), Sensitivity of Hurst parameter estimation to periodic signals in time series and filtering approaches, *Geophys. Res. Lett.*, 32, L17401, doi:10.1029/2005GL024069.

[R3] Nagarajan, R., & Kavasseri, R. G. (2005). Minimizing the effect of periodic and quasi-periodic trends in detrended fluctuation analysis. *Chaos, Solitons & Fractals*, 26(3), 777-784.

[R4] Cuesta, Helena. "Automatic structure detection and visualization in symphonic music." (2015).

[R5] Maddage, N. C. (2006). Automatic structure detection for popular music. *Ieee Multimedia*, 13(1), 65-77.

[R6] Serra, J., Müller, M., Grosche, P., & Arcos, J. L. (2014). Unsupervised music structure annotation by time series structure features and segment similarity. *IEEE Transactions on Multimedia*, 16(5), 1229-1240.

REVIEWERS' COMMENTS

Reviewer #1 (Remarks to the Author):

I note that the authors have gone to great lengths to improve the manuscript by including more illustrations and valuable sections of text that make the manuscript more consistent and easier to read. I believe that the manuscript improved essentially and so I only remain with a few additional comments:}

1) It seems that the length of the plateau shows “an increasing trend in dependence on the lengths of the compositions”. Is to say, the longer the musical piece the longer the duration of the plateau, as nicely shown in Fig. 7. It is possible that the cut-off time is more a direct measure of the length of the piece of music than an indicator of the interplay between anticipation and uncertainty. As mentioned in my previous post, the fact that the power spectrum of musical pieces resembles a power law with exponents between minus one and two is commonly understood as a fine-tuned balance between predictability and surprise. Stronger autocorrelations are usually considered too monotonic, weaker ones too irregular. Both extremes seem to disturb the aesthetic perception. I struggle to see how an additional length scale, where autocorrelations are completely lost, could contribute positively to this $1/f$ fine tuning. For example, if there were a correlation between beta values and cut-off times, so that too strong autocorrelations were compensated for by shorter cut-off times, this could be seen as a strong indication that would support the authors' hypothesis. However, as Fig. 10 of the Supplementary Material shows, such a correlation does not exist. I think it is worth discussing this issue and showing why this additional element is beneficial alongside the balanced autocorrelation function corresponding to the $1/f$ spectra. Besides, up to know the fact that power spectra of musical pieces is close to that of pink (or red) noise is not touched in the manuscript. But due to the fact that an essential part of the interpretation of the authors goes in this direction I consider this aspect as relevant.

2) As already mentioned in the former review, a power spectrum of any shape, even not a $1/f$ spectrum, is an indicator for noise. I think, phrases like “So in a sense one can speak of $1/f$ -noise in musical pitch sequences (and of very slowly decaying correlations),...”, (page 4 second paragraph) should be corrected in the entire text.

3) Just below Eq. (10), the definition of D should be $[-\pi, +\pi) - (-W, +W)$. Is that correct? Otherwise, it is not consistent with the correction of the integration borders of the above formula. Am I right?

Minors:

1) Page 2, paragraph that initiates mentioning the work of Voss and Clark, why $1/f$ -noise is written in bold face?

2) There is a typing error in the caption of Fig. 4 "...down the the bandwidth..."

Reviewer #2 (Remarks to the Author):

Authors have respond to all comment precisely and by details.

REPLY TO REVIEWERS' COMMENTS

We would like to thank the reviewers for their renewed careful reading of our manuscript. We made all changes as suggested by reviewer #1 and as detailed below. We marked the suggestions and suggested corrections of typing errors in *italic*.

Reviewer #1 (Remarks to the Author):

I note that the authors have gone to great lengths to improve the manuscript by including more illustrations and valuable sections of text that make the manuscript more consistent and easier to read. I believe that the manuscript improved essentially and so I only remain with a few additional comments:}

1) It seems that the length of the plateau shows “an increasing trend in dependence on the lengths of the compositions”. Is to say, the longer the musical piece the longer the duration of the plateau, as nicely shown in Fig. 7. It is possible that the cut-off time is more a direct measure of the length of the piece of music than an indicator of the interplay between anticipation and uncertainty. As mentioned in my previous post, the fact that the power spectrum of musical pieces resembles a power law with exponents between minus one and two is commonly understood as a fine-tuned balance between predictability and surprise. Stronger autocorrelations are usually considered too monotonic, weaker ones too irregular. Both extremes seem to disturb the aesthetic perception. I struggle to see how an additional length scale, where autocorrelations are completely lost, could contribute positively to this $1/f$ fine tuning. For example, if there were a correlation between beta values and cut-off times, so that too strong autocorrelations were compensated for by shorter cut-off times, this could be seen as a strong indication that would support the authors' hypothesis. However, as Fig. 10 of the Supplementary Material shows, such a correlation does not exist. *I think it is worth discussing this issue and showing why this additional element is beneficial alongside the balanced autocorrelation function corresponding to the $1/f$ spectra. Besides, up to know the fact that power spectra of musical pieces is close to that of pink (or red) noise is not touched in the manuscript. But due to the fact that an essential part of the interpretation of the authors goes in this direction I consider this aspect as relevant.*

We had discussed this issue already in the third paragraph of the discussion section and are now discussing it more explicitly there.

2) As already mentioned in the former review, a power spectrum of any shape, even not a $1/f$ spectrum, is an indicator for noise. I think, *phrases like “So in a sense one can speak of $1/f$ -noise in musical pitch sequences (and of very slowly decaying correlations),...”, (page 4 second paragraph) should be corrected in the entire text.*

Done; as suggested we clarified these phrases there (due to a change in format now on page 3) and in the last sentence of the section on classical music scores.

3) Just below Eq. (10), the definition of D should be $[-\pi, +\pi] - (-W, +W)$. Is that correct? Otherwise, it is not consistent with the correction of the integration borders of the above formula. *Am I right?*

Yes, we corrected it and replaced $\frac{1}{2}$ by π .

Minors:

1) Page 2, paragraph that initiates mentioning the work of Voss and Clark, why $1/f$ -noise is written in *bold face*?

Corrected

2) There is a *typing error* in the caption of Fig. 4 "...down the the bandwidth..."

Done; should have been: "down to the bandwidth" now figure # 3. (The previous Fig. 1 is now in the supplementary information).